# Analysis of Online Platforms' Free Trial Strategies for Digital Content Subscription

Li Chen [ID]

Broadwell College of Business and Economics, Fayetteville State University, Fayetteville, NC 28301, USA; lchen@uncfsu.edu

**Abstract:** Adopting subscription models has become a popular trend in e-commerce. Digital content platforms often use free trials to attract potential subscribers. However, rigorous research on platforms' free trial strategies is lacking. Our paper aims to fill this gap using a horizontal duopoly framework. We investigate both the scenario of market symmetry and of market asymmetry. We study the benchmark case with no free trial for each scenario and derive equilibrium results with free trial promotion. We find that the platform will not benefit from free trials unless consumers' learning rate exceeds a threshold. When a free trial is introduced in market symmetry, a higher rate of consumers' learning leads to a higher price of subscription and a shorter free trial length. In addition, when the network effect is incorporated, the platform will introduce a lower subscription price and a shorter free trial length. Under market asymmetry, our results show that the platform faces a higher threshold of consumers' learning rate. Additionally, it will offer a higher price and a longer period of free trial compared with those under the market symmetry scenario. Overall, our study provides useful managerial insights for online content platforms when considering their subscription strategies.

**Keywords:** subscription; free trial; consumers' learning rate; market asymmetry; network effect; duopoly

## 1. Introduction

In recent years, the subscription model has become a popular trend in e-commerce [1]. Different from the traditional business model where consumers buy products and services when they need, consumers of the subscription model receive the products and services on a scheduled basis such as monthly or quarterly by signing up for companies' subscription programs [2]. For online platforms of digital content, several well-known examples of subscriptions are online video streaming services (e.g., Netflix and Hulu), audiobooks (e.g., Audible and Audiobook), online music (e.g., Apple iTunes and Spotify), online newspapers (e.g., Wall Street Journal and New York Times), etc. According to a recent consulting report from McKenzie, 46% of more than 5000 people surveyed have subscribed to digital media [3].

Adopting the subscription model turned out to be a win-win decision for both consumers and online platforms of digital content [2]. From the aspect of consumers, they benefit from enjoying digital goods and services at a lower cost and from more accessible sources. They can renew their membership on a regular basis and cancel their subscription at their convenience. For online platforms, they can generate predictable income with their subscription programs, which helps them develop a sustainable long-term business model. In addition, subscription as a business model demonstrated great resilience during the COVID-19 pandemic [4]. Working-from-home arrangements during the COVID-19 pandemic enabled people to spend more time on digital content. Not surprisingly, there was a boosting engagement in subscription programs. For example, Netflix, an online streaming platform, expanded to more than 190 countries in seven years [5]. According to

an industry report from Statista, the number of Netflix subscribers exceeded 220 million in the second quarter of 2022 [6].

To attract more customers, many online platforms of digital content (hereinafter "platforms") introduce a free trial period in an effort to promote subscriptions. To be more precise, they often offer programs that allow users to try their subscriptions for free for a certain period of time. For example, Hulu and AMC Plus have introduced a seven-day free trial for potential consumers who are interested in their online streaming services. YouTube TV and Adobe Creative Cloud offer 14-day free trial programs for their users. Amazon Prime, Economist, and Apple Music offer 30-day free trials for consumers in their subscription programs.

While the free trial strategy is often considered an efficient way to generate sales, it has limitations, and its promotion effect will not be fully exploited without careful design [7]. Consequently, platforms desire to better understand the competition dynamics with or without free trial programs. In particular, they want to find out whether it is beneficial to introduce free trial programs. Also, if they decide to introduce a free trial promotion, how should they determine the optimal length of the free trial and the optimal regular price of the subscription after the free trial comes to an end? However, there is a lack of rigorous research on platforms' strategy of free trial subscription, let alone providing useful insights. Our study aims to fill the gap in this strategic issue. To be more specific, we seek to answer the following research questions:

> *Research question 1: Under what conditions should a digital content platform introduce a free trial promotion?*

> *Research question 2: If a digital content platform chooses to introduce a free trial for promotion, how will the market conditions affect its decisions such as the optimal regular price of the subscription after free trial and the optimal length of free trial?*

> *Research question 3: How will market asymmetry influence platforms' decisions on free trial promotion such as the optimal regular price of the subscription after the free trial and the optimal length of free trial?*

To address these research questions, we employ a duopoly model in a horizontal framework with one platform offering free trial promotion and the other not. We investigate the platform's free trial strategy in both the scenario of market symmetry and the scenario of market asymmetry. For each scenario, we first present the results of the benchmark case where there is no free trial promotion. Then we derive platforms' optimal decisions in terms of prices and free trial length at equilibrium when the free trial promotion is introduced.

Our main findings are as follows. First, it is not always beneficial for platforms to introduce free trial programs. To be more precise, we derive a threshold of consumers' learning rate. The platform will be better off after introducing a free trial promotion only when consumers' learning rate exceeds that threshold. Second, when a free trial is introduced in market symmetry, the platform's optimal price increases with consumers' learning rate, but its optimal free trial length decreases with consumers' learning rate. In addition, our findings show that the platform's optimal regular price and free trial length decrease with the network effect. Third, in the scenario of market asymmetry, we find that the platform will experience a higher threshold of consumers' learning rate. In addition, it will introduce a higher price of subscription and a longer free trial length. Such an effect is more pronounced when there is a higher level of market asymmetry.

The rest of the paper is organized as follows. Section 2 provides a comprehensive literature review of relevant research. In Section 3, we investigate two scenarios: market symmetry and market asymmetry with our horizontal model. For each scenario, we analyze the benchmark case with no free trial and the case where free trial promotion is adopted. We calculate results at equilibrium and derive relevant propositions for the platform. Section 4 discusses both the theoretical insights and the managerial insights based on our findings. Section 5 concludes the paper with future research directions.

## 2. Literature Review

Our study is closely related to two research streams. The first stream is the literature on subscription models for distributing digital goods and services. One group of studies focuses on finding the optimal design of the subscription programs. Refs. [8,9] analyzed the dynamic pricing strategy for subscription providers. Ref. [10] investigated how to make optimal joint decisions regarding subscription price and advertising space for online platforms. In [11], the authors studied an online subscription platform's optimal pricing when consumers' transportation cost is incorporated. Ref. [12] analyzed the optimal balance between subscription fees and commission fees. Ref. [13] examined platforms' optimal subscription pricing strategies under social influence. Ref. [14] explored the optimal planning of subscriptions for an online media platform when consumers have different usage rates. While these studies help platforms derive optimal pricing, they have not considered the impact of free trial promotion programs.

Another group of studies examines the consumers' attitudes towards subscription models. For example, Ref. [15] used a questionnaire of 505 consumers to study their willingness to pay for a music subscription in the presence of online piracy. They reported that online music piracy will negatively influence consumers' willingness to pay. Using a survey of 1356 UK customers, Ref. [16] described the profile of consumers who are most likely to be engaged in the subscription. Based on data on Canada cable TV diffusion, Ref. [17] found that the advertising cost and the subscription price have a significant impact on consumers' decisions to subscribe. Ref. [18] reported that subscription leads to a significant increase in consumers' following purchases after controlling the self-selection effect. Ref. [19] showed that online platforms can use co-opetition strategies to attract subscribers. While this group of research provides general insights into consumers' attitudes and decisions on subscription to digital content, little attention has been paid to the free trial strategy.

The second stream is related to the literature on free trial promotion. Ref. [20] found that a free trial is an effective promotion tool that increases consumers' purchasing behavior in the future. Based on their online experiment, Ref. [21] reported that free trial increases the conversion rate of freemium users. Ref. [22] explored free trial in a multiple-product setting. For digital goods and services, free trial promotion is frequently adopted to attract potential consumers of software [23,24], online publishers' content [25], interactive digital television service [26], online mobile applications [27], mobile data usage service of telecommunications [28], etc. In general, prior research found that free trials can reduce consumers' uncertainty about the quality of the digital goods and services they are interested in, and positively influence their purchase decisions [29,30].

However, if not designed appropriately, a free trial will do harm, rather than benefit, the firms [31]. For example, prior research showed that free trial might bring negative effects due to challenges of retaining consumers [32], cannibalization effect of consumers' adopting free version of digital services [27], and lower efficiency compared with advertisement [33]. Also, Ref. [34] reported that the promotion tool pay-what-you-want (PWYW) could yield higher promotion revenue than a free trial.

Consequently, many researchers focus on exploring whether it will be more profitable for firms to introduce free trial promotion and what are the key elements of their free trial strategy [25,30]. For example, Ref. [30] examined the trade-off between reduced consumer uncertainty and cannibalization of demand. Ref. [25] found that consumers' learning rate must be significantly high for a monopoly software vendor to offer a free trial and the network effect has no impact on the vendor's free trial strategy.

Our paper differs from those studies in the following aspects. First, while they focused on the setting of monopoly, we studied the platforms' free trial strategy under a duopoly framework. So far little research on free trials has been carried out on the competition scenario. Second, we have comprehensively investigated critical factors of the platforms' free trial strategy. For example, we have incorporated the factor of market asymmetry, which is understudied in prior literature. Third, we focus on the industry of digital content subscription while previous researchers only looked at the software industry.

### 3. Model

In this study, we adopt a two-stage duopoly model in the horizontal setting. The duopoly framework is often used in research on digital goods and services, including electronic books [35], cloud computer services [36], software products [37], online video platforms [19], and media platforms with user-generated content [38]. In addition, the duopoly model can reflect the fierce market competition between platforms of digital content subscription. For example, industry reports reflect that there is an intense race between digital content platforms such as the race among Spotify, Apple Music, and Amazon in the online music streaming market [39] and the race of Netflix versus Disney Plus in the online movie streaming market [40]. Ref. [16] also reported that it is a big challenge to attract consumers with a churn rate of 70% because of the increased competition between digital content platforms.

We assume that one platform (platform A) offers free trial promotion while the other platform (platform B) does not. This is consistent with the business practice we observe. For example, in the online movie streaming market, while some platforms such as Hulu provide a 7-day free trial subscription, some other platforms such as Netflix do not offer any free trial program. When a free trial is employed, we apply a two-stage framework, which is often used in literature [33,41]. To be more precise, the whole-time length [0, *m*] is divided into two stages: Stage 1 [0, *rm*] in which platform A offers a free trial and platform B offers a regular price subscription, and Stage 2 [*rm*, *m*] in which both platforms offer a regular price of subscription. The two key variables for platform A are the ratio of free trial length *r* and platform A's regular price of subscription after free trial *pA*. Figure 1 below presents the two-stage framework of our study.

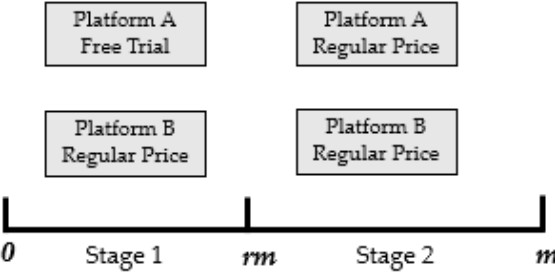

**Figure 1.** Two-stage framework (platform A offers free trial in stage 1).

Following the prior literature, we use several commonly used assumptions such as linear consumer utility function and zero marginal cost of information goods. In addition, we assume a uniform distribution of consumers' preferences between [0, 1] along the axis. Without loss of generality, we set that platform A is located at zero (0) and platform B is located at one (1). Consumers have unfit cost *k* for each unit distance from the platform they prefer. Consumers' intrinsic value *v* from the digital content subscription is high enough so the market is fully covered, which is a common assumption in the literature [36,42]. Table 1 summarizes the notation used in the model.

**Table 1.** Summary of notation.

| Notation | Definition |
|:---:|:---|
| $\theta$ | Location on the horizontal axis |
| $k$ | Intensity of location preference |
| $\beta$ | Intensity of network effect |
| $r$ | Ratio of free trial length over whole time length |
| $m$ | Whole time length |
| $d$ | Measure of market asymmetry |
| $p_A, p_B$ | Price of subscription program for platform A and platform B |
| $\pi_A, \pi_B$ | Revenue for platform A and platform B |

In the current study, we studied both the scenario of market symmetry and the scenario of market asymmetry. For both scenarios, we first present the benchmark case where no online platform of digital content offers a free trial program and then examine the impact of free trial subscription promotion in the market.

### 3.1. Market Symmetry

We start with the platform's free trial in a symmetric market, which refers to the scenario under which consumers' unfit cost $k$ is the same for the two platforms in the market [35]. Duopoly with market symmetry fits the market setting in which competition happens between digital content platforms of similar size such as Spotify versus Apple iTunes in the online music streaming market. We do not discuss the case of both platforms offering free trials because this Bertrand-type competition will lead to an equilibrium in which the two platforms evenly share the market size and charge prices of zero.

#### 3.1.1. Benchmark Case: No Free Trial in Market Symmetry

When the market is fully covered, all consumers use subscriptions to digital content from one of the two platforms. In the absence of platform A's free trial program, consumers who choose a subscription to platform A will derive utility $u_A = v - p_A - k\theta$, and consumers who choose a subscription of platform B will derive utility $u_B = v - p_B - k(1 - \theta)$. Thus, the indifference point $\theta = \frac{k - p_A + p_B}{2k}$. Then we obtain the equilibrium results and receive the following Lemma:

**Lemma 1.** *When there is no free trial in market symmetry, the price of platform A and revenue of platform A at equilibrium are as follows: $p_A^* = k$, $\pi_A^* = \frac{mk}{2}$; The price of platform B, and revenue of platform B at equilibrium are as follows: $p_B^* = k$, $\pi_B^* = \frac{mk}{2}$. (Please see the proof in Appendix A).*

#### 3.1.2. Case 1: Free Trial in Market Symmetry

In this subsection, we study the case in which platform A offers a free trial in the market symmetry. Now we assume that the ratio of free trial length $r$ is fixed and is heterogeneously determined, which helps us focus on the impact of critical factors such as consumers' learning rate and network effect. We relax this assumption later in this section to derive the optimal free trial length ratio. Figure 2 below reflects the horizontal model of market symmetry.

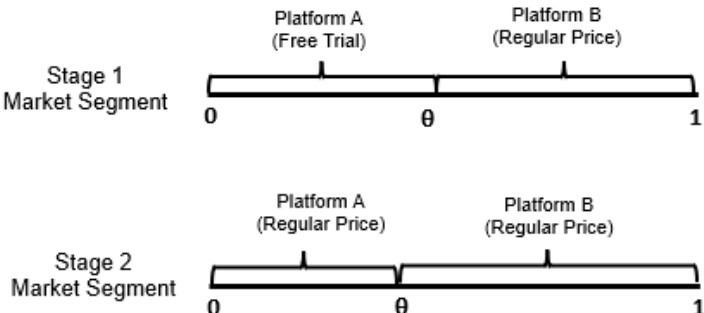

**Figure 2.** Horizontal model of market symmetry.

For stage 1 (free trial from 0 to $rm$, $0 < r < 1/2$), consumers choosing platform A will derive utility $u_A = v - k\theta$, and consumers choosing platform B will derive utility $u_B = v - p_B - k(1 - \theta)$. Thus, we obtain $\theta = \frac{k + p_B}{2k}$. We set $0 < r < 1/2$ because the free trial length is usually shorter in practice. The two-time-period model with the same length period [33] does not fit the online subscription market.

For stage 2 (from $rm$ to $m$), the free trial ends. Consumers choosing platform A will derive utility $u_A = v + s * r * m - p_A - k\theta$, where $s$ refers to the consumers' learning rate of the subscription. Prior research suggests that a time-limited free trial helps consumers increase their valuation when their initial valuation is under-estimated [30]. If consumers overestimate the valuation of the subscription, the platform is unlikely to offer a free trial. Therefore, following the literature, we focus on the case of consumers underestimating the value.

In this stage, consumers choosing platform B will derive the same utility, so $\theta = \frac{k + p_B - p_A + s}{2k}$. Combining these two stages, we obtain $\pi_A = \left(\frac{k - p_A + p_B + s*r*m}{2k}\right) p_A (1 - r) m$ and $\pi_B = \frac{m p_B \left(k + p_A (1-r) - p_B - mrs + mr^2 s\right)}{2k}$. Using the same approach as Lemma 1, we obtain the equilibrium results and the following Lemma:

**Lemma 2.** *When platform A offers free trial promotion in market symmetry, platform A's optimal price and revenue at equilibrium are as follows: $p_A^* = \frac{3k + mrs + mr^2 s}{3+r}$, $\pi_A^* = \frac{m(1-r)(3k + mr(1+r)s)^2}{2k(3+r)^2}$; platform B's optimal price and revenue at equilibrium are as follows: $p_B^* = \frac{3k - kr - mrs + mr^2 s}{3+r}$, $\pi_B^* = \frac{m(k(3-r) - m(1-r)rs)^2}{2k(3+r)^2}$. (Please see the proof in Appendix A).*

Prior research argued that the consumers' learning rate is a key factor of consumers' decisions on experience goods [29,42]. Thus, we compare platform A's revenue at equilibrium of both case 1 and the benchmark case and identify the conditions of consumers' learning rates $s$ for the platform to benefit from introducing its free trial program. Based on the above results, we derive the following proposition.

**Proposition 1.** *(When to introduce free trial)*
*Let $T = \left(\frac{k}{m(r+r^2)}\right)\left(\sqrt{\frac{(3+r)^2}{(1-r)}} - 3\right)$. Platform A will offer a free trial only when consumers' learning rate on subscription is higher than a threshold $s > T$. Under this condition, platform A will obtain a higher revenue at equilibrium than that of the benchmark case. (Please see the proof in Appendix A).*

Based on Proposition 1, we can tell that it is not always optimal for platforms to introduce free trials. It is only profitable when consumers' learning rate of valuation s exceeds the threshold $T$ defined above. This finding might explain why some platforms such as online streaming platform HBO Max do not offer free trial programs. When the quality of digital content subscriptions is widely and clearly known to the general public, consumers' learning rate might not be high enough to benefit the platform. One recommendation based on this finding is that it is crucial for platforms to accurately estimate consumers' learning rate before deciding whether or not to introduce free trials.

Figure 3 below illustrates the value of the threshold of consumers' learning rate $T$ with respect to unfit cost $k$. We can see that the threshold value increases with the unfit cost. In addition, for a given unfit cost $k$, a smaller $m$ corresponds to a higher threshold.

Then we investigate the effect of consumers' learning rates on platform A's decisions: (1) optimal price of subscription after free trial and (2) optimal ratio of free trial length. Lemma 2 presents the value of the optimal price. For the length of the free trial, we relax the assumption that it is heterogeneously given. Following the approach proposed by [30], we first treat the ratio of free trial length as given so the problem reduces to the previous case of solving the optimal price as a function of the ratio of free trial length $r$. Then we take the derivative of the optimal price over the ratio $r$ to compute the optimal ratio of free trial length ($0 < r < 1/2$). We derive the following proposition in terms of the effect of consumers' learning rates on platforms' free trial decisions.

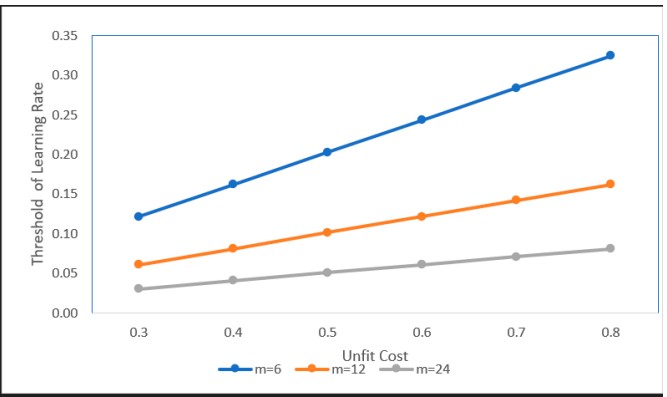

**Figure 3.** Threshold of consumers' learning rate with respect to unfit cost (*r* = 0.1).

**Proposition 2.** *(The effect of consumers' learning rate on platform A's decisions)*
*Platform A's optimal price of subscription increases with consumers' learning rates s. In addition, platform A's optimal ratio of free trial length decreases with consumers' learning rates s. Therefore, a larger consumer learning rate leads to shorter free trials. Note that the condition k > m\*s is necessary to derive a positive optimal free trial length ratio r. (Please see the proof in Appendix A).*

Based on Proposition 2, we can see that consumers' learning rates *s* play a crucial role in platforms' free trial strategy. After a platform decides to introduce a free trial, it can take advantage of higher consumers' learning rates to charge a higher price for its subscription programs. This is because a higher learning rate leads to elevated consumers valuation, which enables the platform to charge a higher price. Figure 4a below shows the platform's optimal price with respect to consumers' learning rate.

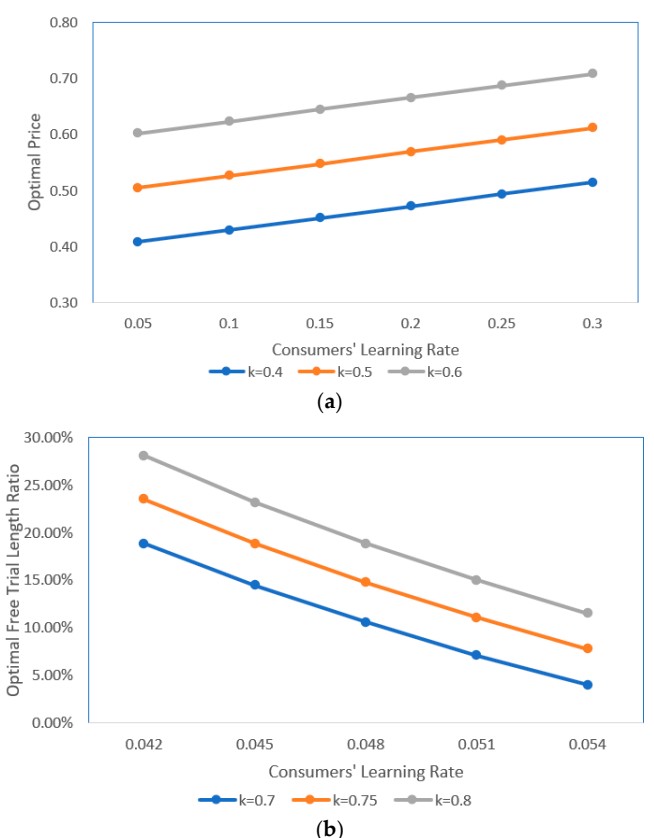

**Figure 4.** (**a**). Optimal price with respect to consumers' learning rates (*m* = 12, *r* = 0.1). (**b**). Optimal free trial length ratio with respect to consumers' learning rates (*m* = 12).

In addition, it is optimal for the platform to offer a shorter free trial length providing that consumers learn faster. The rationale here is that the platform can make use of consumers' faster learning to condense the time length of paid subscription of its digital content for more revenue. Figure 4b below shows the platform's optimal free trial length ratio *r* with respect to consumers' learning rates. We can see that the optimal free trial length will decrease with consumers' learning rates.

3.1.3. Case 2: Free Trial in Market Symmetry with Network Effect

Then we investigate the impact of the network effect in our model. Digital content subscriptions exhibit a network effect [43] like software [24], cloud computing services [44], and online mobile applications [45]. Consumers might benefit from network effects from multiple aspects. First, a large user base can reduce consumers' learning costs because they may find instructions more easily from other users of the same subscription. Second, a large user base is more likely to reduce the setup and maintenance cost per user for the platform. Lastly, the platform is more willing to provide better technical support and customer service for a larger number of subscribers. Consequently, consumers' utility increases $(\beta D)$ where $\beta$ represents the network effect intensity and $D$ is the market share (i.e., the total number of subscribers) [43].

When the network effect is incorporated, for stage 1, consumers choosing platform A will derive utility $u_A = v - (k - \beta)\theta$, with extra utility $\beta\theta$. Consumers choosing platform B will derive utility $u_B = v - p_B - k(1 - \theta) + \beta(1 - \theta)$, with extra utility $\beta(1 - \theta)$. For stage 2, consumers choosing platform A obtain $u_A = v + s * r * m - p_A - (k - \beta)\theta$. Consumers choosing platform B still obtain $u_B = v - p_B - k(1 - \theta) + \beta(1 - \theta)$. Using the same approach as Lemma 1, we obtain the following Lemma for the equilibrium results:

**Lemma 3.** *In the presence of network effect, when platform A offers free trial promotion, platform A's optimal price and revenue at equilibrium are as follows:* $p_A^* = \frac{3k - 3\beta + mrs + mr^2s}{3 + r}$, $\pi_A^* = \frac{m(1-r)(3(k-\beta) + mr(1+r)s)^2}{2(k-\beta)(3+r)^2}$, $r^* = \frac{-3ms + \sqrt{3}\sqrt{kms - \beta ms + 2m^2s^2}}{ms}$; *platform B's optimal price and revenue at equilibrium are as follows:* $p_B^* = \frac{3k - 3\beta - kr + \beta r - mrs + mr^2s}{3 + r}$, $\pi_B^* = \frac{m((k-\beta)(3-r) - m(1-r)rs)^2}{2(k-\beta)(3+r)^2}$. *Note that the condition of $\beta < k$ is necessary for the existence of the optimal solution. (Please see the proof in Appendix A).*

Based on the results of Lemma 3, we are able to study the impact of network effect on platform A's decisions of optimal price and optimal free trial length. Following the approach of Proposition 2, we derive the following proposition:

**Proposition 3.** *(The impact of network effect on platform A's free trial decisions)*
*In the presence of network effect, platform A offers a lower price of subscription compared with the case of no network effect. The stronger the network effect is, the larger the decrease in the price will be. In addition, platform A offers a shorter free trial length compared with the case of no network effect. However, the stronger the network effect is, the smaller the decrease in the free trial length will be. (Please see the proof in Appendix A).*

The results of this proposition describe the impact of the network effect on platform A's free trial strategy. In the presence of a network effect, the platform will offer a lower price and a shorter free trial stage, which implies that the network effect might bring a cannibalization effect on the free trial. Since consumers will benefit from the large user base, the extra utility they receive from the free trial becomes less significant. Figure 5a below shows platform A's optimal price with respect to network effect intensity.

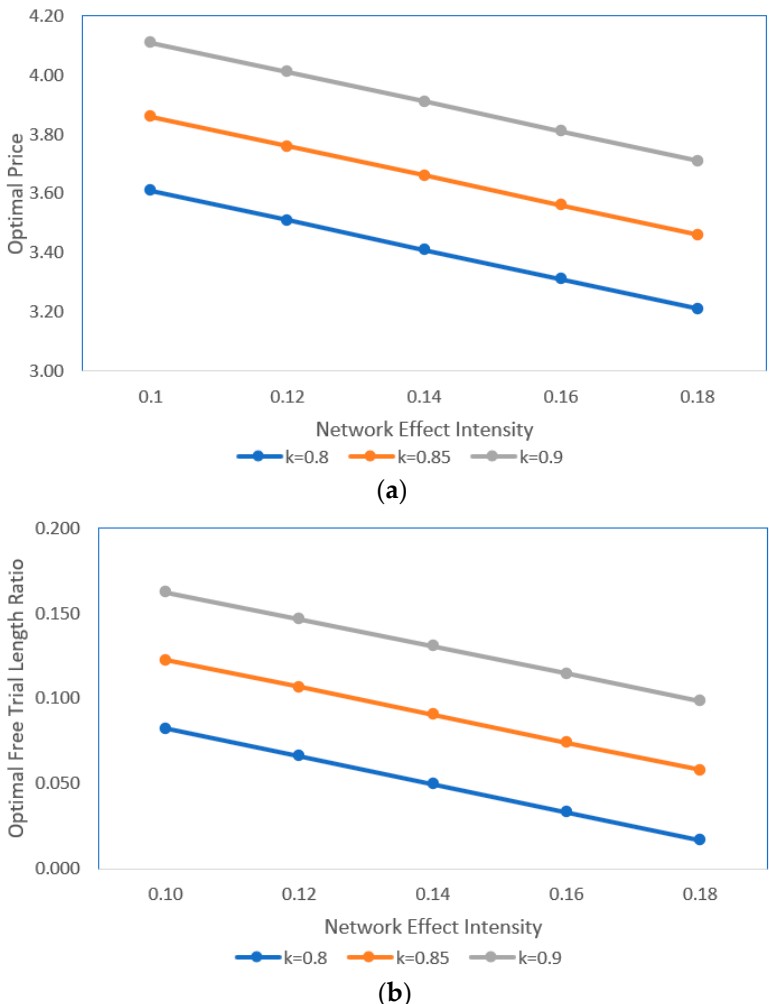

**Figure 5.** (**a**). Optimal price with respect to network effect intensity (*m* = 6, *s* = 0.1). (**b**). Optimal free trial length ratio with respect to network effect intensity (*m* = 6, *s* = 0.1).

Figure 5b below shows platform A's optimal free trial length ratio with respect to network effect intensity. The results clearly reflect that platforms will provide shorter free trials as the network effect intensity increases. This is consistent with our observations. Several online streaming platforms with a large number of subscription users such as Netflix do not offer free trials. For the streaming platform Hulu, which offers a free trial, it only offers a short length (7-day free trial).

*3.2. Market Asymmetry*

In this subsection, we investigate platform A's free trial promotion under the scenario of market asymmetry. Market asymmetry refers to the context in which consumers have different disutility towards the two platforms in the market [24]. We assume that consumers of platform A have a higher level of disutility *d*\**k* (*d* > 1), where *d* denotes the measure of market asymmetry. In other words, everything else being equal, consumers choosing subscription from platform A will incur a higher disutility than those choosing platform B, which will give platform B an advantage.

Market asymmetry is an extension of the conventional horizontal model and better fits a market in which one platform is a dominant player. For example, according to industry reports, Audible took 65% of audiobook sales in the United States in 2022 [46]. Consequently, Audible holds an advantageous position over smaller platforms of audiobooks such as Kobo. Similarly, Spotify took 30.5% of the online music streaming market in the second quarter of

2022 [47] and Netflix took 44.21% of the online movie streaming market in the first quarter of 2023 [48]. Figure 6 below reflects the horizontal model of market asymmetry.

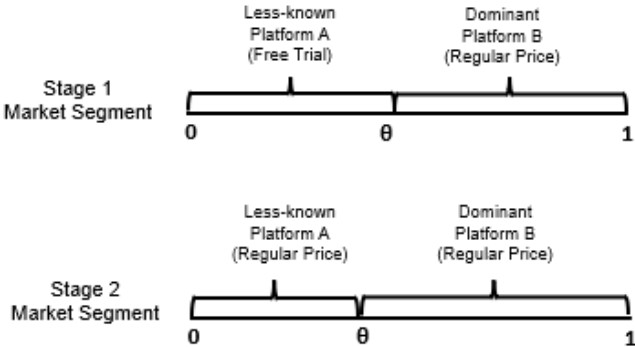

**Figure 6.** Horizontal model of market asymmetry.

### 3.2.1. Benchmark Case: No Free trial in Market Asymmetry

Following the same approach of the benchmark case in market symmetry, we can see that consumers who choose the subscription of platform A will derive utility $u_A = v - p_A - d * k\theta$, and consumers who choose the subscription of platform B will derive utility $u_B = v - p_B - k(1 - \theta)$. Using the same approach as Lemma 1, we obtain the equilibrium results in the following Lemma:

**Lemma 4.** *When there is no free trial in market asymmetry, the price of platform A, and revenue of platform A at equilibrium are as follows: $p_A^* = \frac{(2+d)}{3}k$, $\pi_A^* = \frac{(2+d)^2 mk}{9(1+d)}$; The price of platform B, and revenue of platform B at equilibrium are as follows: $p_B^* = \frac{(1+2d)}{3}k$, $\pi_B^* = \frac{(1+2d)^2 mk}{9(1+d)}$. (Please see the proof in Appendix A).*

### 3.2.2. Case 3: Free Trial in Market Asymmetry

In this subsection, we explore the case of platform A offering a free trial in the scenario of market asymmetry. For stage 1, consumers choosing platform A will derive utility $u_A = v - d * k\theta$, and consumers choosing platform B will derive utility $u_B = v - p_B - k(1 - \theta)$. For stage 2, consumers choosing platform A will derive utility $u_A = v + s * r * m - p_A - d * k\theta$, and consumers choosing platform B will derive the same utility. Using the same approach as Lemma 1, we obtain the following equilibrium results:

**Lemma 5.** *When platform A offers free trial promotion in market asymmetry, platform A's optimal price and revenue at equilibrium are as follows: $p_A^* = \frac{(2+d)k + mrs + mr^2 s}{3+r}$, $\pi_A^* = \frac{m(1-r)((2+d)k + mr(1+r)s)^2}{(1+d)k(3+r)^2}$, $r^* = \frac{-3ms + \sqrt{2kms + dkms + 6m^2 s^2}}{ms}$; platform B's optimal price and revenue at equilibrium are as follows: $p_B^* = \frac{(2+d)k - kr - mrs + mr^2 s}{3+r}$, $\pi_B^* = \frac{m(k(1+2d-r) - m(1-r)rs)^2}{(1+d)k(3+r)^2}$. (Please see the proof in Appendix A).*

By comparing the equilibrium results of both market symmetry and market asymmetry, we are able to investigate the impact of market asymmetry. We first look at its impact on the threshold of consumers' learning rate based on which platforms will decide whether to introduce a free trial or not. Using the same approach as Proposition 3, we obtain the following proposition:

**Proposition 4.** *(The impact of market asymmetry on threshold)*
*In the scenario of market asymmetry, platform A will have a higher threshold of consumers' learning rate to introduce free trial compared with the threshold in the scenario of market symmetry. The higher the measure of market asymmetry is, the larger the threshold of the consumers' learning rate. (Please see the proof in Appendix A).*

The results of Proposition 4 reflect that market asymmetry has an incremental effect on the threshold for the platform to introduce free trial programs. In the scenario of market asymmetry, consumers of platform A need to have higher utility to offset the initial disadvantage. If consumers have a low learning rate, it is highly likely that their utility after the free trial stage still remains low, which makes it unnecessary to provide free trials. In other words, there is a narrower space for platform A to introduce free trials. This might explain why many smaller online streaming services are inactive to provide their own free trial programs. Figure 7 below reflects the threshold of consumers' learning rates with respect to the measure of market asymmetry.

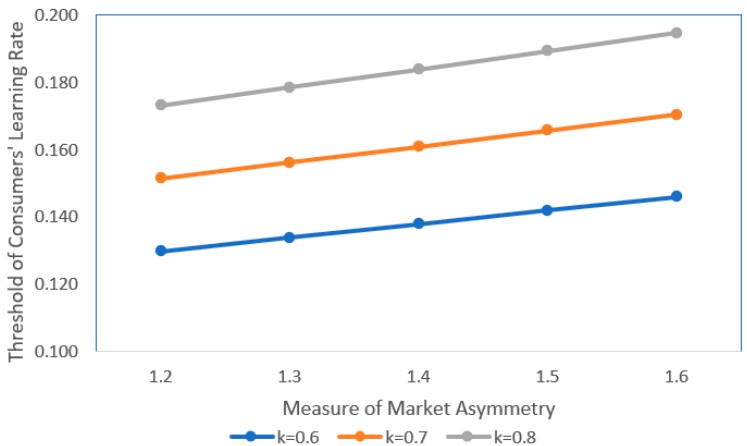

**Figure 7.** Threshold of consumers' learning rates with respect to market asymmetry ($m = 12$, $r = 0.1$).

Then we examine the impact of market asymmetry on the platform's free trial decisions including both the optimal price of the subscription and the optimal free trial length. Using the same approach as Proposition 3, we derive the following proposition:

**Proposition 5.** *(The impact of market asymmetry on free trial decisions)*
*In the scenario of market asymmetry, platform A offers a higher price of subscription compared with the scenario of market symmetry. The stronger the market asymmetry effect is, the larger the increase in the price. In addition, platform A offers a longer free trial length compared with the scenario of market symmetry. Similarly, the stronger the market asymmetry effect is, the larger the increase in the free trial length. (Please see the proof in Appendix A).*

Based on the results of Proposition 5, we can see that market asymmetry will lead to higher prices and a longer free trial length. Our finding of higher optimal price implies that there is actually less intense price competition in the scenario of market asymmetry. Compared with the scenario of market symmetry in which consumers have the same preference level towards platforms, platforms under market asymmetry already differentiate themselves in the aspect of consumers' preferences. Consequently, there is less pressure to carry out intense price competition to attract consumers. Figure 8a below compares the optimal price in both the scenario of market symmetry and the scenario of market asymmetry with respect to the measure of market asymmetry.

In addition, our finding shows that platform A needs to offer a longer free trial in the scenario of market asymmetry for consumers to increase their valuation of its subscription service. The higher the level of market asymmetry is, the longer the free trial length compared with the scenario of market symmetry will be. One recommendation for the platform is that it needs to wait longer for consumers to improve their evaluation during the free trial. Figure 8b below compares the optimal free trial length ratio in both the scenario of market symmetry and the scenario of market asymmetry.

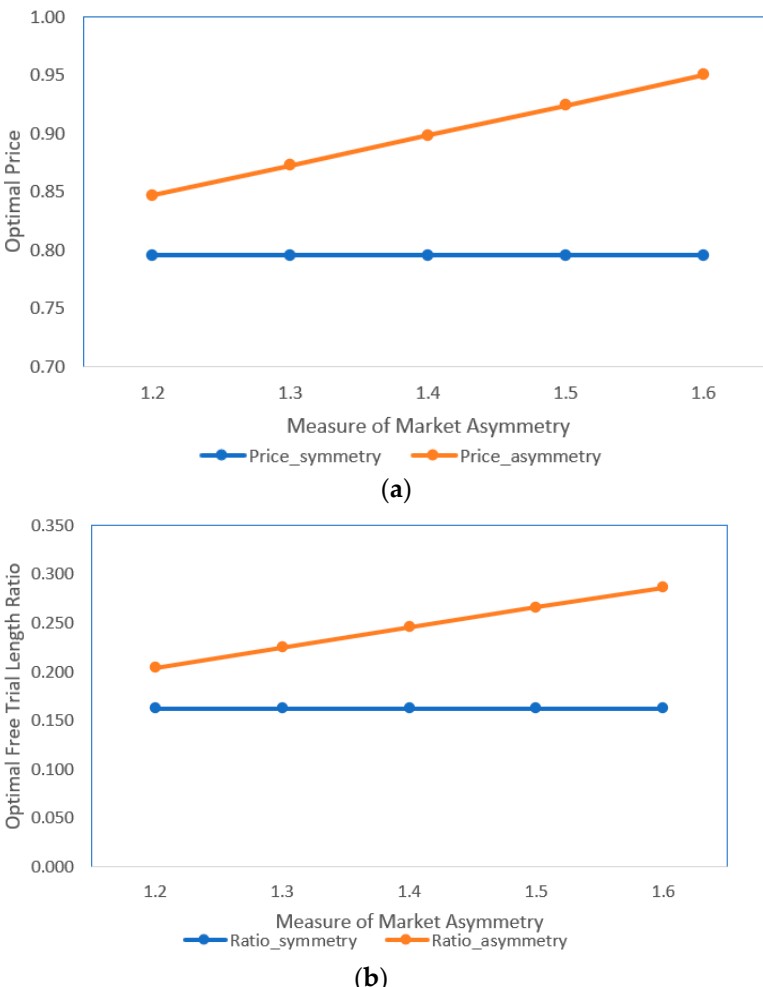

**Figure 8.** (**a**). Comparison of optimal price (*m* = 6, *s* = 0.2, *k* = 0.8 and *r* = 0.1), (**b**). Comparison of optimal free trial length ratio (*m* = 6, *s* = 0.2, and *k* = 0.8).

## 4. Theoretical and Managerial Implications

### 4.1. Theoretical Implications

The theoretical implications of our paper lie in several aspects. First, this study enriches the literature on e-commerce by exploring the impact of platforms' free trial subscription programs. While most prior studies of subscriptions in e-commerce focus on topics such as whether platforms should choose a subscription model [49], the optimal pricing of subscriptions [10], and consumers' attitudes toward subscriptions [15], little research has been carried out on the platforms' promotion strategies such as free trial programs.

Second, we extend the literature on a free trial of digital content by applying a horizontal duopoly framework. So far, most related studies in the literature focus on the monopoly setting of digital goods such as software [30,37]. The scenario of competition between platforms in real business practice is understudied in extant research. Our paper provides theoretical guidance for platforms that plan to implement free trial programs in the competitive market.

Third, this paper examines another dimension in the subscription market: unfit cost setup. We look at two scenarios with different unfit cost setups (market symmetry versus market asymmetry). So far research in this area is scarce. To the best of our knowledge, our study is the first attempt to comprehensively investigate the impact of consumers' learning rate, network effect, and market asymmetry on platforms' free trial strategies.

*4.2. Managerial Implications*

In addition, our study provides managerial implications in the following three aspects. Firstly, we derive the threshold of consumers' learning rates, which helps the platforms determine whether to introduce a free trial subscription or not. In practice, platforms can estimate consumers' learning rate through consumer surveys, industry reports, or historical data from market consulting companies, and determine their free trial strategy accordingly.

Secondly, we have come up with policies for the platforms' optimal subscription price and free trial length. In addition, we discuss the role the network effect has played in platforms' free trial strategy. Based on our findings, platforms are able to decide whether to lower/raise the price or extend/shorten the free trial length in response to changes in the market.

Finally, our study analyzes the case where the platform might be at a disadvantage in the market. In business practice, we find that many platforms that introduce free trials are in a weaker position in competition. By incorporating the factor of market asymmetry, our results provide deeper and richer insights into the platforms' free trial strategies in a dynamic market.

## 5. Conclusions

Using subscription models in the distribution of digital content has become a popular trend in e-commerce. To attract potential consumers, online platforms often use free trials such as 14-day or 30-day free trials to promote their subscription programs. However, there is little research on platforms' free trial strategy. In particular, platforms might face new challenges after taking consumers' learning rate, network effect, and market asymmetry into consideration. Using a two-stage horizontal framework, our study is an exploratory attempt to investigate whether platforms should offer free trial programs and how to determine the optimal length of free trial and the regular price for after-free-trial subscriptions. We aim to provide platforms with useful insights that will help them better develop free trial programs in the market.

Based on a duopoly model in a horizontal framework, we investigate platforms' free trial promotion in two scenarios: market symmetry and market asymmetry. For each scenario, we derive the results at equilibrium and find optimal free trial decisions for platforms in terms of the price of the subscription program and free trial length. For both scenarios, we find that platforms will not introduce a free trial unless consumers' learning rate exceeds a threshold. In addition, our results show that consumers' learning rate has a positive influence on the optimal price of subscription but a negative influence on the optimal free trial length in market symmetry. Our findings also reflect that the network effect in the market will lead to a lower price and a shorter free trial length. Finally, we find that market asymmetry has an incremental effect on the platforms' price and free trial length.

Our research is not without limitations. One limitation is that we focused on one type of promotion—free trial promotion. In practice, we observed that some platforms offer both free trial promotions and discount promotions. For example, the *Economist* offers both a one-month free trial and a 50% discount for its new *Economist* plus online podcast. Another limitation is that our study did not take bundling into account. While some platforms such as Disney offer bundling programs of online streaming services, incorporating the case of bundling will make our model overcomplicated. Finally, we did not include individual contributors such as popular YouTubers' free trial promotions into account because their free trial programs change all the time.

Our study can be extended for future research in the following aspects: First, we focus on a duopoly setting to derive useful managerial insights for platforms of digital content subscription. This market can be more complex and investigating the platforms' strategy in an oligopoly setting might be a promising direction. Second, the platforms in our model only offer one subscription program. Future research may look at the multi-subscription case. For example, Netflix offers three types of programs to its customers:

The basic program costs USD 9.99 per month, the standard program USD 15.49 per month, and the premium program USD 19.99 per month. It might be interesting to study how platforms' free trials will change correspondingly. Lastly, future research might study the free trial strategy in the case of the dual pricing and the mixed pricing system. Ref. [49] proposed mixed pricing combining ownership and subscription. Ref. [50] reported that some firms such as cell phone communication service providers offer a dual pricing strategy of subscription services (a monthly access fee and per minute usage charge). It might be interesting to explore what will happen in those settings.

**Funding:** The research has received no external funding.

**Institutional Review Board Statement:** Not applicable.

**Informed Consent Statement:** Not applicable.

**Data Availability Statement:** No new data were created or analyzed in this study. Data sharing is not applicable to this article.

**Acknowledgments:** The author wants to thank Cliff Nowell of Weber State University and Jianhui Peng of Shanghai Normal University for their valuable input and support in my research.

**Conflicts of Interest:** The author declares no conflict of interest.

**Appendix A**

**Proof of Lemma 1.** (Symmetric Market, Benchmark Case)

We have $u_A = v - p_A - k\theta$ and $u_B = v - p_B - k(1 - \theta)$. Letting $u_A = u_B$, we obtain $\theta = \frac{k - p_A + p_B}{2k}$. Thus, the revenue of two competing platforms are as follows:

$$\pi_A = m\left(\frac{k - p_A + p_B}{2k}\right)p_A$$

$$\pi_B = m\left(1 - \frac{k - p_A + p_B}{2k}\right)p_B$$

Solving the first-order conditions (FOCs) ($\frac{\partial \pi_A}{\partial p_A} = 0$ and $\frac{\partial \pi_B}{\partial p_B} = 0$), we obtain equilibrium results:

$$p_A^* = p_B^* = k; \ \theta = \frac{1}{2}; \ \pi_A^* = \pi_B^* = \frac{mk}{2}$$

$\square$

**Proof of Lemma 2.** (Case 1: Symmetric Market, Free Trial Case)

For stage 1, platform A offers a free trial so $u_A = v - k\theta$, and $u_B = v - p_B - k(1 - \theta)$. Letting $u_A = u_B$, $\theta = \frac{k + p_B}{2k}$. For stage 2, there is no free trial so $u_A = v + s * r * m - p_A - k\theta$, where $s$ refers to consumers' learning rate. Since $u_B$ remains the same, we obtain $\theta = \frac{k + p_B - p_A + s * r * m}{2k}$. So $\pi_A = \left(\frac{k - p_A + p_B + s * r * m}{2k}\right)p_A(1 - r)m$ and $\pi_B = \frac{mp_B(k + p_A(1 - r) - p_B - mrs + mr^2 s)}{2k}$. Using the same approach as Lemma 1, we obtain the following equilibrium results:

$$p_A^* = \frac{3k + mrs + mr^2 s}{3 + r}, \ \pi_A^* = \frac{m(1 - r)(3k + mr(1 + r)s)^2}{2k(3 + r)^2};$$

$$p_B^* = \frac{3k - kr - mrs + mr^2 s}{3 + r}, \ \pi_B^* = \frac{m(k(-3 + r) - m(-1 + r)rs)^2}{2k(3 + r)^2}$$

$\square$

**Proof of Proposition 1.** (When to Introduce Free Trial)

We calculate the difference of revenue in Case 1 and benchmark case:

$$\Delta \pi_A = \frac{m(1-r)(3k+mr(1+r)s)^2}{2k(3+r)^2} - \frac{mk}{2}$$

To obtain $\Delta \pi_A > 0$, we need $s > T = \left(\frac{k}{m(r+r^2)}\right)\left(\sqrt{\frac{(3+r)^2}{(1-r)}} - 3\right)$. $\square$

**Proof of Proposition 2.** (Effect of Consumers' Learning Rate)

For the optimal price $p_A^* = \frac{3k+mrs+mr^2s}{3+r}$, $\frac{\partial p_A}{\partial s} = \frac{mr(1+r)}{3+r} > 0$.

To compute the optimal length of the free trial, we need to have $\frac{\partial p_A}{\partial r} = \frac{m(3+6r+r^2)s-3k}{(3+r)^2} = 0$.

$$r^* = \frac{\sqrt{3}\sqrt{kms+2m^2s2}}{ms} - 3$$

For the optimal length of the free trial, $\frac{\partial r}{\partial s} = -\frac{\sqrt{3}k}{2s\sqrt{ms(k+2ms)}} < 0$. $\square$

**Proof of Lemma 3.** (Case 2: Symmetric Market, Free Trial with Network Effect)

For stage 1, $u_A = v - (k-\beta)\theta$, and $u_B = v - p_B - (k-\beta)(1-\theta)$. Thus, $\theta = \frac{k-\beta+p_B}{2(k-\beta)}$. For stage 2, $u_A = v + s*r*m - p_A - k\theta + \beta\theta$, and $u_B = v - p_B - k(1-\theta) + \beta(1-\theta)$. We obtain $\theta = \frac{k-\beta+p_B-p_A+s*r*m}{2(k-\beta)}$. Thus, $\pi_A = \left(\frac{k-\beta+p_B-p_A+s*r*m}{2(k-\beta)}\right)p_A(1-r)m$ and $\pi_B = mp_B\left(1 - \frac{k-\beta+p_B}{2(k-\beta)}\right)r + mp_B(1-r)\left(1 - \frac{k-\beta-\text{pa}+\text{pb}+mrs}{2(k-\beta)}\right)$. Using the same approach as Lemma 1, we obtain the following equilibrium results:

$$p_A^* = \frac{3k-3\beta+mrs+mr^2s}{3+r}, \pi_A^* = -\frac{m(-1+r)(3k-3\beta+mr(1+r)s)^2}{2(k-\beta)(3+r)^2}$$

$$r^* = \frac{-3ms+\sqrt{3}\sqrt{kms-\beta ms+2m^2s2}}{ms}$$

$$p_B^* = \frac{3k-3\beta-kr+\beta r-mrs+mr^2s}{3+r}, \pi_B^* = \frac{m((k-\beta)(3-r)+m(-1+r)rs)^2}{2(k-\beta)(3+r)^2}$$

$\square$

**Proof of Proposition 3.** (Impact of Network Effect)

To compare the optimal price of case 1 and case 2, we obtain

$$\Delta p_A^* = \left(\frac{3k-3\beta+mrs+mr^2s}{3+r}\right) - \left(\frac{3k+mrs+mr^2s}{3+r}\right) = -\frac{3\beta}{3+r} < 0$$

In addition, $\frac{\partial \Delta p_A^*}{\partial \beta} = -\frac{3}{3+r} < 0$.

Using the same approach of Proposition 2, $\frac{\partial p_A}{\partial r} = \frac{ms(1+2r)}{3+r} + \frac{-3k+3\text{nbeta}-mrs-mr^2s}{(3+r)^2} = 0$.

$$r^* = \frac{\sqrt{3}\sqrt{(k-\beta)ms+2m^2s2}}{ms} - 3$$

$$\Delta r^* = \left(\frac{\sqrt{3}\sqrt{(k-\beta)ms+2m^2s2}}{ms} - 3\right) - \left(\frac{\sqrt{3}\sqrt{kms+2m^2s2}}{ms} - 3\right) < 0$$

In addition, $\frac{\partial \Delta r^*}{\partial \beta} = -\frac{\sqrt{3}}{2\sqrt{ms(k-\text{nbeta}+2ms)}} < 0.$ $\square$

**Proof of Lemma 4.** (Asymmetric Market, Benchmark Case)

We have $u_A = v - p_A - d*k\theta$ and $u_B = v - p_B - k(1-\theta)$. Letting $u_A = u_B$, we obtain $\theta = \frac{k - p_A + p_B}{(d+1)*k}$. Thus, the revenue of two competing platforms are as follows:

$$\pi_A = m\left(\frac{k - p_A + p_B}{(d+1)*k}\right)p_A$$

$$\pi_B = m\left(1 - \frac{k - p_A + p_B}{(d+1)*k}\right)p_B$$

Solving the first-order conditions (FOCs) ($\frac{\partial \pi_A}{\partial p_A} = 0$ and $\frac{\partial \pi_B}{\partial p_B} = 0$), we obtain equilibrium results:

$$p_A^* = \frac{(2+d)k}{3}; p_B^* = \frac{(1+2d)k}{3}; \pi_A^* = \frac{mk((2+d))^2}{9(1+d)}; \pi_B^* = \frac{mk(1+2d)^2}{9(1+d)}$$

$\square$

**Proof of Lemma 5.** (Asymmetric Market, Free Trial Case)

For stage 1, $u_A = v - d*k\theta$, and $u_B = v - p_B - k(1-\theta)$. Letting $u_A = u_B$, $\theta = \frac{k+p_B}{(1+d)k}$. For stage 2, $u_A = v + s*r*m - p_A - d*k\theta$. Since $u_B$ remains the same, we obtain $\theta = \frac{k+p_B-p_A+s*r*m}{(1+d)k}$. Using the same approach as Lemma 1, we obtain the following equilibrium results:

$$p_A^* = \frac{(2+d)k + mrs + mr^2 s}{3+r}, \pi_A^* = \frac{m(1-r)((2+d)k + mr(1+r)s)^2}{(1+d)k(3+r)^2};$$

$$r^* = \frac{-3ms + \sqrt{2kms + dkms + 6m^2 s^2}}{ms}$$

$$p_B^* = \frac{(2+d)k - kr - mrs + mr^2 s}{3+r}, \pi_B^* = \frac{m(k(1+2d-r) - m(1-r)rs)^2}{(1+d)k(3+r)^2}$$

$\square$

**Proof of Proposition 4.** (Impact of Market Asymmetry on Threshold)

We calculate the difference of revenue in Case 3 and benchmark case:

$$\Delta\pi_A = \frac{m(1-r)((2+d)k + mr(1+r)s)^2}{(1+d)k(3+r)^2} - \frac{(2+d)^2 km}{9(1+d)}$$

To obtain $\Delta\pi_A > 0$, we need $s > T_A = \frac{(3+r)(2+d)k}{3mr(1-r)}\sqrt{\frac{1}{(1-r)}} - \frac{2k+dk}{m(r+r^2)}$.

In addition, let $w = \frac{1}{mr(1-r)}$, $\frac{\partial T_A}{\partial d} = \frac{k\left(3mr(1+r^2) + \frac{(2+d)k(3+r)^2}{k(3+r)(2+d)w}\right)}{3m^2(1-r)r^2(1+r)^2} > 0$.

We compare the threshold of consumers' learning rate in market symmetry and market asymmetry. Let $w = \frac{1}{mr(1-r)}$, we obtain:

$$\Delta T = \left(\frac{1}{3}\sqrt{\frac{(3+r)^2(2+d)^2}{(1-r)}} - \sqrt{\frac{(3+r)^2}{(1-r)}} - (d-1)\right)wk > 0$$

$\square$

**Proof of Proposition 5.** (Impact of Market Asymmetry on Platform's Decision)

Under market symmetry, $p_A^* = \frac{3k+mrs+mr^2s}{3+r}$.

Under market asymmetry, $p_A^* = \frac{(2+d)k+mrs+mr^2s}{3+r}$.

Taking the difference, we obtain $\Delta p_A^* = \frac{(d-1)k}{3+r}$. In addition, $\frac{\partial \Delta p_A^*}{\partial d} = \frac{k}{3+r} > 0$.

Under market symmetry, $r^* = \frac{\sqrt{3}\sqrt{kms+2m^2s^2}}{ms} - 3$.

Under market asymmetry, $r^* = \frac{\sqrt{2kms+dkms+6m^2s^2}}{ms} - 3$.

Taking the difference, we obtain $\Delta r^* = \frac{-\sqrt{3}\sqrt{ms(k+2ms)}+\sqrt{ms((2+d)k+6ms)}}{ms} > 0$. In addition, $\frac{\partial \Delta r^*}{\partial d} = \frac{k}{2\sqrt{ms((2+d)k+6ms)}} > 0$. $\square$

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
