# Peer review of "Analysis of Online Platforms’ Free Trial Strategies for Digital Content Subscription"

_jtaer, doi:10.3390/jtaer18040106_

Round 1

Reviewer 1 Report

Comments and Suggestions for Authors

Congratulations for the good work regarding the impact of platform's free trial subscription!

The paper presents a relevant perspective of waht e-commerce platforms should do and when.

For the purpose of improving the article I recommend the following:

- Introduction and literature review need mosre supportive literature. Even tough all the aspects are very well placed and presented, the sections still needs more recent references;

- The methodology should dtart wit the tehory intoriduction. I is presented but more references are needed in the first part of this section;

- The paper needs  the introduction of the objective/s. It/they must be recalled in the discussion section;

- Try to create an overall image to illustrate the framework/methodology flow;

- Try to explain better why market assymetry is a relevant concept for the study;

- Conclusions must present the scientific novelty, managerial implications, acadamic implications, limitations of the study, and future research directions.

Reviewer 2 Report

Comments and Suggestions for Authors

Dear Authors: This paper is interesting with good efforts. But why it is lacking in the creation of interest, for example, no framework explained, no model figure? No effort to explain the research design (A diagram). By putting some effort, into the topic due to its maximum use in consumer science, it will create interest. 

1. Abstract: Two sentences, edited. The abstract is an important part of the manuscript. 

However, rigorous research on platforms’ free trial strategies is lacking. We study the benchmark case with no free trial for each scenario and derive equilibrium results with free trial promotion. 

Introduction: Recently, the subscription model has become a popular trend in e-commerce. (First sentence suggestion). model of subscription or subscription model?

The introduction has many mane paragraphs, please merge them if applicable. 

2. The second heading is the literature review.  Third is the "model." Need yo revise the names. To create the interest. 

3. Model Needs one framework. (Diagram)

3. Many serial citations. 

4. Page 12 has first and second. Page 11 also has the first and second. This is confusing. In one heading, two times, two different classifications, but with the same ....

Can change the page 12th first second with firstly and secondly etc. 

5.  Authors need to define the main focused variables. Articles have this lack point. Need efforts.

6. No diagram about the methods or the main research design. This paper has the potential but lacks the reader's interest. (Due to no framework explanation etc. in the figure)

7. Page 11: Line 446-448: 33,7,17. Please avoid repeating the citations. Need to check the complete file.

8. Page 12: Line 505, is this word necessary to write in bold style?

9. This manuscript has only  33 ref. for the 16 pages. Need to check the complete file about the need for citations, and some from this journal. To increase the interest. 

10. Clearly need to define, market symmetry, and subscription e-commerce with an example, in clear terms,  duopoly framework,.

In discussion and a few more places, few general statements, for example, our study contributes to... Lines 452-453, need to edit.

Comments on the Quality of English Language

Need to recheck by an English expert. 

Reviewer 3 Report

Comments and Suggestions for Authors

The theme of the paper is very interesting.

The study is current and well founded.

But, I think that information is still needed on the part of the current bibliography.

On the other hand, the part of conclusions needs to be developed a lot. It must be clear what the implications of the study are, including the managerial implications.

I recommend publishing the paper after these changes.

Reviewer 4 Report

Comments and Suggestions for Authors

Thank you for the work. I found it as highly significant and professional. I don't have any comments.

Good luck.

Author Response

Thank you very much for your encouragement and comments. We believe we have addressed all of the concerns from the other three reviewers. The manuscript is revised accordingly. Hope you will find the improved manuscript publishable in JTAER.

Round 2

Reviewer 2 Report

Comments and Suggestions for Authors

The author's work quality is acceptable, and the work is productive in the scholarly field. Congratulations. 

Comments on the Quality of English Language

I didn't find any major issues, but need to make sure that the English language has no mistakes, etc.